# Frequency-Domain Reverse-Time Migration with Analytic Green’s Function for the Seismic Imaging of Shallow Water Column Structures in the Arctic Ocean

**DOI:** 10.3390/s23146622

**Published:** 2023-07-23

**Authors:** Seung-Goo Kang, U Geun Jang

**Affiliations:** 1Division of Earth Sciences, Korea Polar Research Institute, Incheon 21990, Republic of Korea; ksg9322@kopri.re.kr; 2Department of Geological Sciences, Chungnam National University, Daejeon 34134, Republic of Korea

**Keywords:** analytical Green’s function, frequency-domain reverse-time migration, full-band frequency seismic imaging, seismic oceanographic imaging, thermocline structures in the shallow water column in the Arctic Ocean

## Abstract

Seismic oceanography can provide a two- or three-dimensional view of the water column thermocline structure at a vertical and horizontal resolution from the multi-channel seismic dataset. Several seismic imaging methods and techniques for seismic oceanography have been presented in previous research. In this study, we suggest a new formulation of the frequency-domain reverse-time migration method for seismic oceanography based on the analytic Green’s function. For imaging thermocline structures in the water column from the seismic data, our proposed seismic reverse-time migration method uses the analytic Green’s function for numerically calculating the forward- and backward-modeled wavefield rather than the wave propagation modeling in the conventional algorithm. The frequency-domain reverse-time migration with analytic Green’s function does not require significant computational memory, resources, or a multifrontal direct solver to calculate the migration seismic images as like conventional reverse-time migration. The analytic Green’s function in our reverse-time method makes it possible to provide a high-resolution seismic water column image with a meter-scale grid size, consisting of full-band frequency components for a modest cost and in a low-memory environment for computation. Our method was applied to multi-channel seismic data acquired in the Arctic Ocean and successfully constructed water column seismic images containing the oceanographic reflections caused by thermocline structures of the water mass. From the numerical test, we note that the oceanographic reflections of the migrated seismic images reflected the distribution of Arctic waters in a shallow depth and showed good correspondence with the anomalies of measured temperatures and calculated reflection coefficients from each XCDT profile. Our proposed method has been verified for field data application and accuracy of imaging performance.

## 1. Introduction

Observing the spatial and temporal states of the ocean is critical in understanding global and regional ocean dynamics. In traditional physical oceanography, measurement equipment, such as eXpendable Bathymetry Thermography (XBT) or Conductivity Temperature Depth (CTD), was used to measure the physical oceanographic parameters for each discrete survey point target. Typical automated arrays of sensors, such as the ARGO float system, can collect larger datasets over a long time but still offer only vertical point coverage. The emerging field of seismic oceanography has shown that two-dimensional multi-channel seismic data can provide unprecedented and continuous spatial oceanographic information horizontally with high resolution [1,2]. A two-dimensional seismic image of the water column, built from the multi-channel seismic dataset, contains the reflection events caused by the differences in the physical oceanographic parameters in each water layer. The seismic sections of the water layer can be used to image the water column thermocline structures and understand the physical oceanographic status in spatial states.

Generally, traditional reflection seismic data processing steps and procedures were used to construct the seismic oceanographic images. Smillie [3] described seismic data processing methods that could image water column structures using data sorting, normal moveout correction, stacking, migration, and other conventional reflection seismic imaging techniques. Despite the advanced state of seismic reflection data processing techniques, seismic oceanographic imaging for shallow water column structures remains challenging due to the limited acoustic reflectivity of water masses [4], which generally yields low signal-to-noise ratios.

Reverse-time migration (RTM) is the preferred and most commercially available tool for seismic imaging in structurally complex geological settings. This method identifies correct depth locations for reflectors by assuming a known smooth background model [5] and using an expensive but accurate solution to the two-way wave equation. The frequency-domain RTM algorithm contains the source estimation and source deconvolution steps [6]. These methods are suitable for minimizing bubble oscillations and improving the quality of the results in early-time signals. However, due to its computation and memory requirements, frequency-domain RTM has not been used extensively in imaging water column structures. Given that the scale of most water column reflectors is approximately 10 m or less, handling frequencies must be higher than ~75 Hz to construct the seismic migrated image of the thermocline structures in the water column. Memory requirements and computational cost for detecting features of this scale in conventional frequency-domain RTM preclude using a multifrontal solver [7] for calculating the forward- and backward-modeled wavefield. However, assuming that the sound speed and density within the water column are constant, analytic Green’s functions can be used instead of numerically calculating the forward/backward wavefield using wave propagation modeling in the RTM method. This approach uses incommensurably less memory, has a lower computational cost than the conventional frequency-domain migration methods mentioned above, and can construct meter-scale grid size, high-resolution migrated water column images with over 75 Hz frequency components.

In this study, we first review the formulation of the conventional frequency-domain RTM based on wave propagation modeling using the finite element method and then reformulate the RTM equations using an analytic Green’s function for constructing the water column seismic images. Our proposed algorithm aims to image the thermocline structures in the shallow water column without high computational requirements. The proposed algorithm was tested on multi-channel seismic and XCTD data acquired in the Arctic region simultaneously to verify the field data application. We confirmed that the reformulated RTM based on analytic Green’s function could construct seismic oceanographic images from multi-channel seismic data, and the reflection events in the migrated oceanographic images at shallow water depths correspond well with the seismic reflection profiles calculated by the XCTD data. As a result of the numerical test for the field data application, we note that the analytic Green’s function can be applied to the frequency domain RTM for imaging the water column structures at shallow depths without high-performance computation. The seismic oceanographic images constructed by our proposed algorithm contain full-band frequency components of the seismic field dataset, which can be used to observe the fine thermocline structure in the water column for understanding ocean circulations in the Arctic. We expect our proposed RTM method to be used as a practical imaging tool for seismic oceanographic imaging, especially in observing a fine thermocline structure under mixed layers.

## 2. Materials and Methods

### 2.1. Conventional Virtual Source Imaging Conditions for Frequency-Domain RTM

RTM in the time domain can be expressed as a zero-lag cross-correlation between the partial derivative wavefield with respect to subsurface geophysical parameters such as P-wave velocity, density or acoustic impedance, and time series field dataset [6]:(1)$$\phi_{k}=\int_{0}^{T_{max}}{\left[\frac{\partial u\left(t\right)}{\partial m_{k}}\right]}^{T}d\left(t\right)dt,$$
where $\phi_{k}$ indicates the two-dimensional migration image for the model parameter $m_{k}$, $T_{max}$ is the recording length of the dataset, $\frac{\partial u\left(t\right)}{\partial m_{k}}$ is the partial derivative wavefield, and $d\left(t\right)$ is the field dataset [6]. In the frequency domain, RTM can be obtained by cross-correlation of the Fourier transform partial derivative wavefield with respect to the model parameter $m_{k}$ and the Fourier transform seismic data [6,8]. This formula can be expressed in the frequency domain as
(2)$$\phi_{k}=\int_{0}^{\omega_{max}}\phi_{k}\left(\omega \right)d\omega =\int_{0}^{\omega_{max}}R\left\{{\left[\frac{\partial u\left(\omega \right)}{\partial m_{k}}\right]}^{T}d^{*}\left(\omega \right)\right\}d\omega ,$$
where ω is the angular frequency, d(ω) is the Fourier transform data, $\frac{\partial u\left(\omega \right)}{\partial m_{k}}$ is the partial derivative wavefield, T is the matrix transpose operator, * is the conjugate operator, and R indicates the real part of a complex value [6]. The direct calculation of partial derivative wavefields can be avoided by using virtual source methods.

Assuming a known acoustic background model, the modeled wavefield $u\left(\omega \right)$ can be obtained by solving the Helmholtz equation:(3)$$\nabla \cdot \left(\frac{1}{\rho \left(x\right)}\nabla u\left(\omega \right)\right)+\frac{\omega^{2}}{\rho \left(x\right)c^{2}\left(x\right)}u\left(\omega \right)=-\frac{s\left(\omega \right)}{\rho \left(x\right)}\delta \left(x-x_{s}\right),$$
where $c\left(x\right)$ and $\rho \left(x\right)$ are the sound speed and density of media, respectively, $s\left(\omega \right)$ is the source, and $\delta \left(x-x_{s}\right)$ is the Kronecker delta function representing the point source for a marine seismic survey. By the finite element method or finite difference method, Equation (3) can be expressed in matrix form as
(4)$$S\left(\omega \right)u\left(\omega \right)=f,$$
where f is the source vector, and S is the complex impedance matrix [6]. The impedance matrix consists of the stiffness $K_{k}$, damping $C_{k}$, and mass $M_{k}$ matrixes as defined in the finite element method [9]:(5)$$S\left(\omega \right)=\sum_{k=1}^{ne}S_{k}\left(\omega \right)=\sum_{k=1}^{ne}K_{k}+i\omega C_{k}+\omega^{2}M_{k}.$$

The partial derivative wavefield can be calculated by differentiating Equation (4) with respect to the model parameter $m_{k}$:(6)$$\frac{\partial S\left(\omega \right)}{\partial m_{k}}u\left(\omega \right)+S\left(\omega \right)\frac{\partial u\left(\omega \right)}{\partial m_{k}}=0$$

Rearranging (6) gives
(7)$$\frac{\partial u\left(\omega \right)}{\partial m_{k}}=S^{-1}\left(\omega \right)v_{k}\left(\omega \right),$$
where the virtual secondary source $v_{k}$ is defined as
(8)$$v_{k}\left(\omega \right)=-\frac{\partial S\left(\omega \right)}{\partial m_{k}}u\left(\omega \right),$$
or the product of $\frac{\partial S\left(\omega \right)}{\partial m_{k}}$ and the modeled wavefield $u\left(\omega \right)$. In Equation (8), $\frac{\partial S\left(\omega \right)}{\partial m_{k}}$ controls the radiation pattern of the scattered wavefield and can be reduced to $\frac{\partial S\left(\omega \right)}{\partial m_{k}}$ since a model parameter is defined only at each unique element (see Equation (5) and Shin et al. [8]). For example, the partial derivative of the impedance matrix with respect to $c_{k}$ can be expressed as
(9)$$\frac{\partial S\left(\omega \right)}{\partial c_{k}}=\frac{\partial S_{k}\left(\omega \right)}{\partial c_{k}}=\omega^{2}\frac{\partial M_{k}}{\partial c_{k}}.$$

Substituting Equation (7) into Equation (2) and assuming symmetric properties for $S\left(\omega \right)$ in isotropic modeling give
(10)$$\phi_{k}\left(\omega \right)=R\left\{v_{k}^{T}\left[S^{-1}\left(\omega \right)d^{*}\left(\omega \right)\right]\right\}.$$

By convolving the back-propagated data with virtual sources, an RTM image can be obtained.

Conventional RTM typically uses a multifrontal direct solver when solving Equation (10) due to the sparsity of the complex impedance matrix $S\left(\omega \right)$ and multi-shot simulation [10]. Upon factorization of $S\left(\omega \right),$ forward- and back-propagated wavefields can be obtained by forward and backward substitution, respectively. However, acoustic wave propagation modeling using the finite element method has dispersion and pollution issues [11]. For example, the required grid size of approximately 0.00073 km for a frequency of 128 Hz requires an infeasible memory allotment to perform the factorization [7]. This obstacle can be avoided by applying an analytic Green’s function to the RTM.

### 2.2. Green’s Function Application

Assuming that the density and sound speed are constant in the water column, an analytic Green’s function exists for the unbounded medium
(11)$$u\left(\omega \right)=g\left(x_{r},\omega |x_{s}\right)=\frac{e^{i\frac{\omega }{c_{0}}\left|x_{r}-x_{s}\right|}}{4\pi \left|x_{r}-x_{s}\right|}-\frac{e^{i\frac{\omega }{c_{0}}\left|x_{r}-x_{s}^{\prime }\right|}}{4\pi \left|x_{r}-x_{s}^{\prime }\right|},$$
where the first term on the right-hand side of the equation represents the free-space Green’s function resulting from the true source position $x_{s}=\left(x_{s},y_{s},{+z}_{s}\right)$ while the second term represents the free-space Green’s function from an imaginary source position $x_{s}^{\prime }=\left(x_{s},y_{s},{-z}_{s}\right).$ These are interpreted as the direct arrival and free-surface reflection (respectively) when the reflection coefficient is −1 at z=0. Equation (3) can be solved when
(12)$$c\left(x\right)=c_{0},\rho \left(x\right)=1,and\;s\left(\omega \right)=1.$$

The virtual source in Equation (8) can be archived by using the analytic Green’s function from the source position $x_{s}$ to the scattering point $x_{k}$ as
(13)$$v_{k}\left(\omega \right)=-\frac{\partial S\left(\omega \right)}{\partial m_{k}}g\left(x_{k},\omega |x_{s}\right),$$
while the back-propagated wavefield can be calculated by using the analytic Green’s function from the receiver point $x_{r}$ to scattering point $x_{k}$ as
(14)$$S^{-1}\left(\omega \right)d^{*}\left(\omega \right)=d^{*}\left(\omega \right)g\left(x_{k},\omega |x_{r}\right).$$

Substituting Equations (13) and (14) into (10) gives
(15)$$\phi_{k}\left(\omega \right)=R\left\{{\left[-\frac{\partial S_{k}\left(\omega \right)}{\partial m_{k}}g\left(x_{k},\omega |x_{s}\right)\right]}^{T}d^{*}\left(\omega \right)g\left(x_{k},\omega |x_{r}\right)\right\}$$

In contrast to conventional RTM, the approach outlined above avoids dispersion and pollution issues [11]. A grid size of modified RTM can be determined by considering the spatial resolution for the maximum frequency. For example, a required grid size of approximately 0.002 km for 128 Hz is approximately three times larger than the grid size required for the same frequency using the finite element method. Moreover, $\phi_{k}\left(\omega \right)$ can be calculated without solving the matrix Equation (4), thus bypassing the need for factorization [7]. Therefore, this new method can generate useful results without significant memory requirements.

Given the constant-velocity median assumptions, the proposed method cannot handle events such as free-surface multiples or internal multiples. However, in the case of seismic oceanography, only 0.1–0.001% of energy bounces back, and the intensity of these reflections decays in a manner proportional to the square of the distance, even in a lossless medium. In practice, the total loss will be higher due to the attenuation of sound in seawater, multiple scattering, and the roughness of the seawater surface [9]. Under these conditions, the amplitude of multiples at the air-fluid interface becomes much smaller. Thus, in contrast to conventional RTM, multiples exert very little influence on the results generated by the method described above.

### 2.3. Source Deconvolution Application

Bubble oscillations may affect the seismic oceanographic image results in shallow water settings. Source deconvolution can help minimize these effects and enhance the resultant image quality. In the frequency domain, the source deconvolution can be expressed as
(16)$$d^{\prime }\left(x_{r},\omega |x_{s}\right)=\frac{d\left(x_{r,}\omega |x_{s}\right)}{s^{\prime }\left(\omega \right)},$$
where $s^{\prime }\left(\omega \right)$ is the source signature. The source $s^{\prime }\left(\omega \right)$ can represent a measured signal [12] or an estimated source generated by least-square optimization within the frequency domain [6]:(17)$$s^{\prime \left(\omega \right)}=\frac{d^{T}\left(x_{r},\omega |x_{s}\right)g^{*}\left(x_{r},\omega |x_{s}\right)}{g\left(x_{r,}\omega |x_{s}\right)g^{*}\left(x_{r},\omega |x_{s}\right)}.$$

For deep-sea marine seismic data, direct waves including the direct arrival, its sea-surface reflection, and bubble oscillations can be completely separated from the sea-floor reflection. If $d\left(x_{r},\omega |x_{s}\right)$ contains subsurface reflections, $s^{\prime }\left(\omega \right)$ will not provide a good approximation for the desired source.

Time windowing can help remove subsurface reflections [13]. Green’s function $g\left(x_{r,}\omega |x_{s}\right)$ includes terms that refer to direct arrival and free-surface reflections. Bubbles in the observed data will be transferred to the estimated source, and oscillations in the observed data will be canceled out by the deconvolution steps in Equation (16). Substituting Equation (17) into Equation (15) gives
(18)$$\varphi_{k}\left(\omega \right)=R\left\{{\left[-\frac{\partial S_{k}\left(\omega \right)}{\partial m_{k}}g\left(x_{k,}\omega |x_{s}\right)\right]}^{T}{\left[d^{\prime }\left(\omega \right)\right]}^{*}g\left(x_{k},\omega |x_{r}\right)\right\}$$

## 3. Results and Discussion

### 3.1. Description of the Dataset (Multichannel Seismic Data and XCTD Profiles)

We demonstrate our proposed water column imaging method, the RTM algorithm, by applying it to multichannel seismic field data acquired in the Arctic Ocean provided by the Korea Polar Research Institute (KOPRI). Multichannel seismic data were collected on the Mackenzie Trough in the Canadian Beaufort Sea from 31 August to 4 September 2017 during the Arctic expedition of the Korean ice breaker research vessel ARAON (Cruise Name: ARA08C). For the multichannel seismic survey, the air gun array comprised two Sercel generator-injector (G.I.) air guns. The total volume of the air gun source for this survey was 210 cubic inches, and the shot interval was 25 m. One hundred twenty channels, 1.5 km length of the streamer, were used to record the reflected seismic signal, and the sampling rate of the seismic data was one millisecond. Table 1 presents the acquisition parameters for the multichannel seismic survey during ARA08C. During the multichannel seismic survey, 23 XCTD probes were deployed and measured physical oceanographic information on the survey tracks, where the water depths were deeper than 200 m. Table 2 shows detailed information on the XCTD measurements. The locations of seismic tracks and XCTD deployment points in the Mackenzie Trough of the Canadian Beaufort Sea are shown on a map (Figure 1). In this map, the solid white lines indicate the multichannel seismic track lines, and the red circles indicate the locations of the XCTD measurements.

### 3.2. Numerical Test

For the numerical test, the migrated seismic images for the water column structures were constructed by the proposed frequency-domain RTM with an analytic Green’s function from the field dataset. We choose 1430 m/s (see the blue-solid line in Figure 2) as the constant-velocity median assumptions of the water column for seismic imaging by our proposed RTM method as the following result of the numerical analysis for measuring the travel time difference between field-measured sound velocity by the XCTD profiles and several constant velocity cases. In Figure 2, the travel time difference of wave propagation in the water column between the field-measured sound velocity and constant velocities (1420, 1425, 1430, 1435, and 1440 m/s) were presented, and we can confirm that the constant velocity 1430 m/sec (black solid-line in Figure 2) indicates the smallest travel time difference between the field-measured value (The travel time difference is almost 0.0 s under 500-m depth when we use a constant velocity 1430 m/s). It means that the constant sound velocity of 1430 m/s is almost the same as the used field measuring sound velocity and can be used in our proposed RTM method for imaging the shallow water column environment (~750 m depth) in the Arctic Ocean.

In the case of the density, variation of the density in the water column can also be assumed to be constant because the seismic reflection coefficients for the water layer are represented as about 10^−5^ (see the seismic reflections coefficients in Figure 3 and Figure 4b), which is a very small value compared with subsurface cases. 

In this numerical test for the field dataset, frequencies of 3.6 Hz to 96 Hz with 0.2 Hz intervals were used for seismic imaging of the water column, and the grid size of migrated images was 1.5625 m. We applied a one-sided cosine tapper window [15] at 0–0.02 km depth to avoid singularity close to sources and receiver locations. The migrated seismic images were compared with the temperature, salinity, and reflection coefficient profiles calculated from the XCTD data for each point to confirm the accuracy and verify the thermocline structures in the water column.

Figure 3 shows the temperature–salinity profiles and migrated seismic sections with estimated reflection coefficient profiles measured on the seismic track BF05 (see the map in Figure 1). Figure 3a–c show the temperature–salinity profiles and migrated seismic sections with reflection coefficients at the locations of XCTD02, XCTD03, and XCTD04, respectively. Figure 4 shows the temperature–salinity profiles and migrated seismic sections with reflection coefficient profiles for two measurement points on seismic track BF06. Figure 4a,b present the temperature–salinity profiles and migrated seismic sections with the reflection coefficient for the locations at XCTD05 and XCTD06, respectively. The maximum water depth of the measurement locations for XCTD02~06 presented in Figure 3 and Figure 4 was approximately 400 m on seismic tracks BF05 and BF06. The presented seismic sections for each XCTD site, as shown in Figure 3 and Figure 4, contain seismic reflections between 100 and 200 m. These seismic reflection events in the seismic images show the biggest amplitude event, except the mixed layer (usually a thin 5~10 m depth), of the Arctic Ocean [16] and are well matched with the estimated reflection coefficient profiles calculated from XCTD data for each location of the reflectors in the water column (see the gray arrows in the seismic sections in Figure 3 and Figure 4). Reviewing the seismic sections and comparison with XCTD profiles in Figure 3 and Figure 4, we confirmed that our proposed RTM with an analytic Green’s function could construct accurate seismic sections for imaging the water mass boundaries of the Arctic Ocean and thermocline structures of the water column in shallow water depth.

Next, we conducted another numerical test to verify the imaging performance of the proposed frequency-domain RTM method in a more deep-sea environment. Figure 5 presents the migrated seismic image of seismic track BF09 with five XCTD profiles measured on the BF09 seismic track. The total length of seismic section BF09 was 90 km, and the maximum water depth of this survey line was 1.8 km. Figure 5a presents the temperature–salinity profiles for XCTD07, 08, 09, 10, and 11, which were measured sequentially on the BF09 line. Figure 5b shows the migrated seismic sections for the water column at each XCTD measurement site (XCTD07~11) with calculated seismic reflection coefficient profiles. Reviewing each migrated seismic section for each XCTD site proposed in Figure 5b, we note that the seismic reflection events, which have the largest amplitude except for mixed layer (~10 m depth) were imaged at depths of 100 and 200 m (see the gray arrows in Figure 5b). These events were well matched with the XCTD’s reflection coefficient profiles at the same depth. Figure 5c shows the zoomed seismic image for the shallow water column part, presented with the reflection coefficient profiles for each XCTD site; we note the occurrence of seismic events in the thermocline near the 200-m depth, and they well matched the reflection coefficient for each XCTD site.

Following Woodgate [16], the profile of the Arctic Ocean consists of a mixed layer, along with Pacific water, and Atlantic water. The Pacific water in the Arctic is distributed in the shallow depth under the mixed layer; salinity is under 33 PSU. On the other hand, Atlantic water is distributed deeper than 200 m with high salinity and high temperature (over 0 °C). We note that imaged seismic reflections near the 200 m depth of our oceanographic migrated images in Figure 5 indicate the boundary between the Pacific and Atlantic waters in the Arctic circle, and our proposed RTM method successfully images the water mass boundaries in the shallow water column in the Arctic Ocean.

In Figure 6, Figure 7 and Figure 8, the migrated seismic sections, constructed by the frequency-domain RTM based on the analytic Green’s function with the XCTD profiles, were presented, confirming the thermocline structures in the shallow water column part. Figure 6 is the seismic oceanographic image of BF10 for the shallow water depth with the reflection coefficient for each XCTD station. Figure 7 shows the seismic oceanographic image of BF11 for the shallow water depth with the reflection coefficient for the XCTD20 station. Figure 8 presents the seismic oceanographic image of BF12 for shallow water depths with the reflection coefficients for stations XCTD22 and 23. We note that constructed reflections around the 200-m depth in seismic images were well matched with events of the reflection coefficient calculated by the XCTD profiles, as shown in Figure 6, Figure 7 and Figure 8. These seismic oceanographic events also can interpret the boundaries between the Pacific and Atlantic waters in the Arctic Ocean [16], following Woodgate [16].

From the numerical test for the field dataset, we confirm that our proposed frequency-domain RTM with the analytic Green’s function can be used for observing the oceanographic thermocline structures in the water column from the multi-channel seismic dataset under the assumption of a constant sound velocity (1430 m/s) for the water column. Our proposed RTM method is a new seismic imaging technique that remarkably reduces memory usage and computational requirements for seismic imaging. We expect that our RTM method will be a practical seismic oceanographic method in the future.

## 4. Discussion

### 4.1. The Reason Why We Cannot Compare with Conventional Modeling-Based RTM Method

The frequency-domain RTM is an advanced seismic imaging method that is complicated and expensive, but the great advantage of RTM is that it can generate accurate seismic images from multi-channel seismic data. Our proposed RTM method is based on the analytic Green’s function, and it was developed to consider seismic wave propagation in three dimensions. In the case of the conventional frequency-domain RTM, wave propagation modeling is required to calculate the forward and backward wavefield; we must solve the matrix of the wave equation of the FEM for the three dimensions. Seismic oceanography requires a high-frequency band (around 100 Hz) with a meter-scaled grid for water column structure imaging, and it is not possible to FEM matrix factorize and construct migrated images using conventional frequency-domain RTM under the limitational computing resources. There is no way to construct seismic oceanographic images for shallow water column parts using a conventional modeling-based RTM algorithm. The analytic Green’s function makes it possible to avoid a huge amount of computing procedures (FEM matrix factorization) and can handle three dimensions under the small workstation system. Our proposed RTM method based on the analytic Green’s function is the only way to construct the RTM seismic image with a high-frequency component (almost full-frequency band) of the water column in shallow waters.

### 4.2. The Reason Why We Cannot Compare with Conventional Seismic Data Processing Result

Our proposed RTM does not require a preprocessing procedure, such as de-ghosting and de-bubble steps, because our proposed RTM method contains a source estimation algorithm, which works in the de-bubble and de-ghosting steps; we confirmed that our source estimation algorithm works well in our RTM method and presents reasonable seismic oceanographic images without bubble and ghost effects. However, to obtain a seismic image using conventional data processing procedures, acquiring multi-channel seismic data must be performed in several preprocessing steps. The bubbles are a significant obstacle in seismic oceanographic images in the case of shallow water depths; obtaining an interpretable seismic image using a standard seismic imaging process is not easy and not guaranteed, especially in shallow water depths. For that reason, we do not perform conventional standard data processing to obtain a seismic oceanographic image, especially in shallow water depths, and we cannot compare the RTM results with the seismic oceanographic images using conventional data processing procedures.

### 4.3. Discussion about Assuming the Constant P-Wave Velocity and Density of the Water Column for Oceanographic Seismic Imaging by the Analytic Green’s Function

To obtain a seismic oceanographic image using RTM based on the analytic Green’s function, we assume that the P-wave velocity and density of the water column are constant. In Figure 2, the wave propagation travel time difference for the water column between the field-measured sound velocity and constant P-wave velocities (1420, 1425, 1430, 1435, and 1440 m/s) is presented, and we can note that the travel time difference is too small, especially in shallow depths under 750 m. It means that the P-wave velocity of the water column can be assumed to be the constant value for seismic imaging of the water column structures in shallow water depths. Our presented numerical examples (Figure 3, Figure 4, Figure 5, Figure 6, Figure 7 and Figure 8) show that the assumption of constant velocity does not matter in the case of the frequency-domain reverse-time migration for constructing the water column image in shallow water depths.

## 5. Conclusions

Based on the frequency-domain RTM algorithm, we newly formulated a seismic imaging method with the analytic Green’s function for imaging the meter-scaled thermocline structures in the shallow part of the water column. The method can handle higher-frequency bands by reformulating numerical calculation of forward- and back-propagated wavefields using analytic Green’s under the assumption that the density and P-wave velocity of the water column is constant. The analytic Green’s function in our RTM method makes it possible to generate seismic oceanographic images in shallow depths with a meter-scale grid size. However, conventional RTM based on the FEM wave propagation modeling method cannot construct a meter-scale seismic oceanographic images for the shallow part of the water column, because it requires a huge amount of computing processes for FEM matrix factorization. Therefore, directly comparing oceanographic imaging performance with the conventional RTM method is impossible. The numerical test for the field data acquired in the Arctic Ocean demonstrated that our proposed RTM method could generate a high-quality water column image containing the seismic reflections caused by the fine-scaled thermocline structures and boundaries between the Arctic water masses with only modest computational cost and memory requirements. The migrated seismic sections were verified by comparing them with XCTD profiles. As the results of the numerical test, we note that the wave propagation modeling in the frequency-domain RTM can be replaced with the analytic Green’s function for imaging the seismic oceanographic structures from the seismic field dataset and improving computational efficiency. Furthermore, we confirm that constant sound velocity and density can be used for constructing the seismic migration images for the water column case by RTM. We expect our proposed migration method to become a valuable imaging tool for oceanographic research.

## Figures and Tables

**Figure 1 sensors-23-06622-f001:**
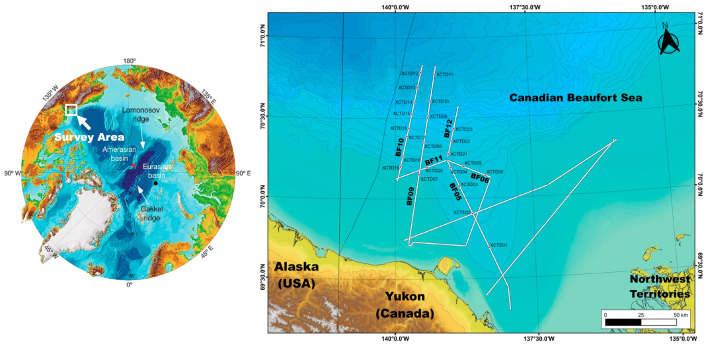
Map of the multichannel seismic tracks and the XCTD measurement stations during the ARA08C expedition on the Canadian Beaufort shelf.

**Figure 2 sensors-23-06622-f002:**
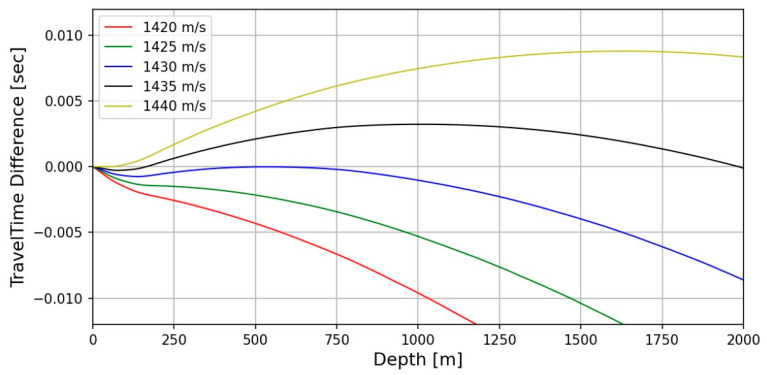
Calculated travel time difference by wave propagation of the water column for sound velocity variation between field-measured and constant values (1420, 1425, 1430, 1435, and 1440 m/s).

**Figure 3 sensors-23-06622-f003:**
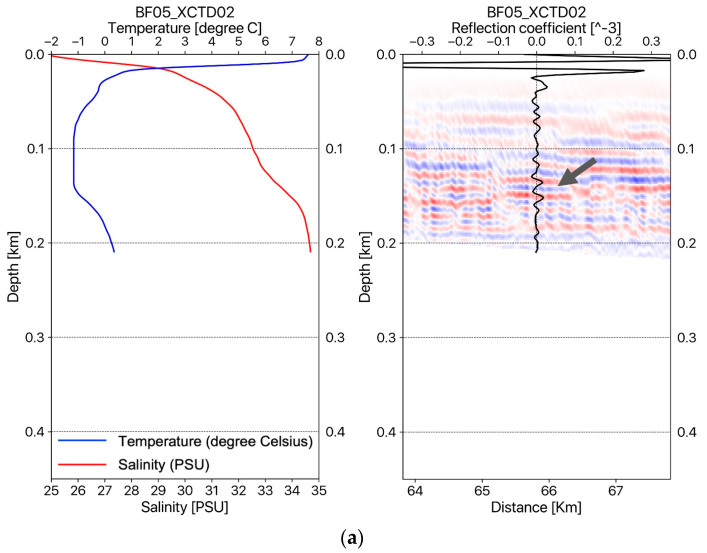
(**a**) The temperature–salinity profile and an oceanographic seismic section for the BF05 line with the reflection coefficient for the location at XCTD02; (**b**) the the temperature–salinity profile and oceanographic seismic section for the BF05 line with the reflection coefficient for the location at XCTD03; (**c**) the the temperature–salinity profile and oceanographic seismic section for the BF05 line with the reflection coefficient for the location at XCTD04.

**Figure 4 sensors-23-06622-f004:**
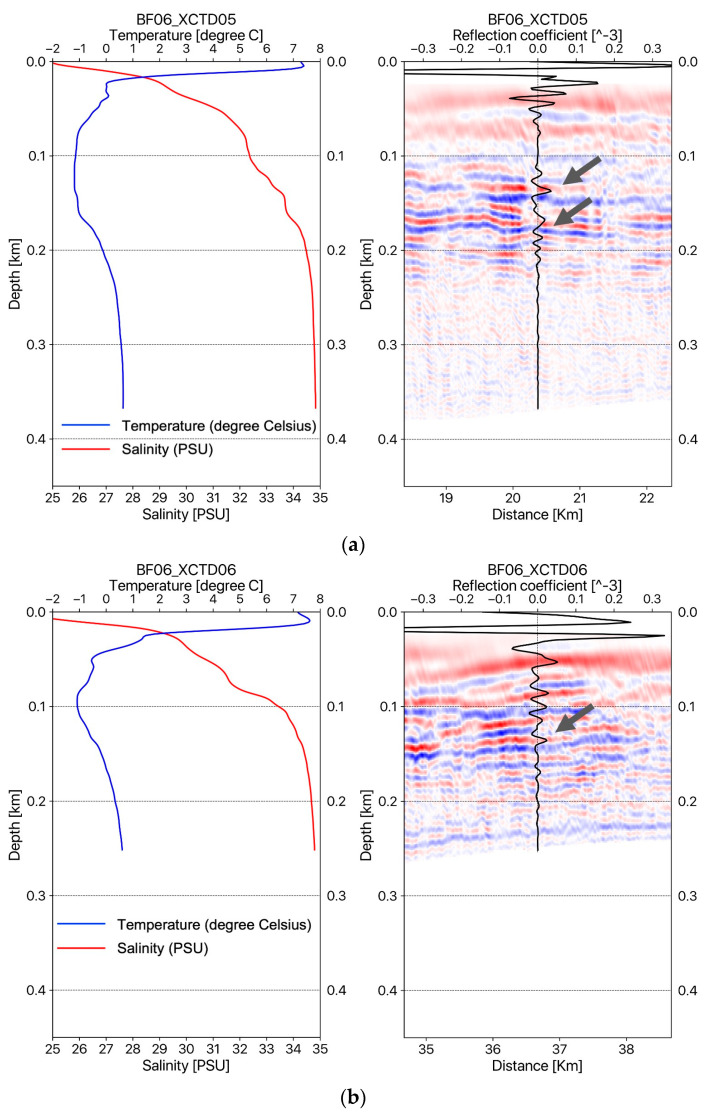
(**a**) The temperature–salinity profile and oceanographic seismic section for the BF06 line with the reflection coefficient for the location at XCTD05; (**b**) the temperature–salinity profile and oceanographic seismic section for the BF06 line with the reflection coefficient for the location at XCTD06.

**Figure 5 sensors-23-06622-f005:**
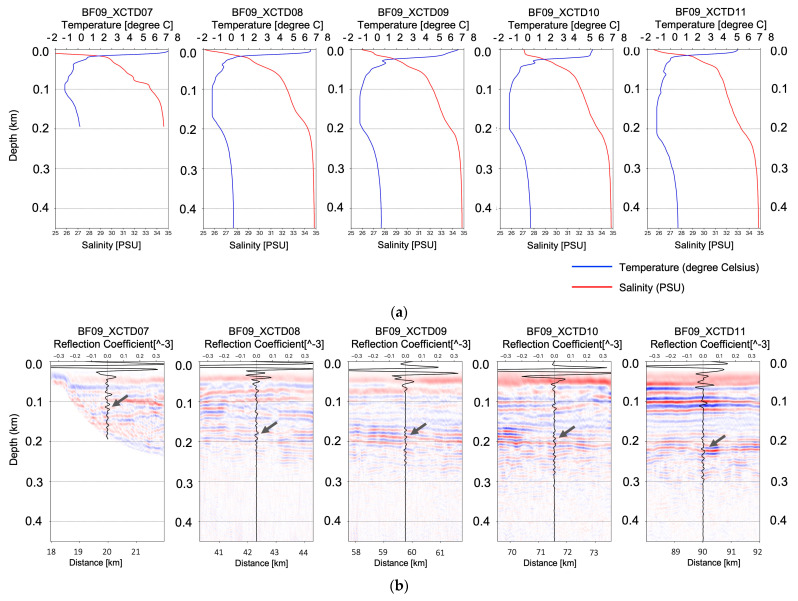
(**a**) The temperature–salinity profiles of XCTD07~11; (**b**) the oceanographic seismic sections of the BF09 line for each XCTD station (XCTD07~11) with estimated reflection coefficients; (**c**) the zoomed seismic oceanographic image of the BF09 for the shallow water depth (under 300 m) with the reflection coefficient profiles for each XCTD station.

**Figure 6 sensors-23-06622-f006:**
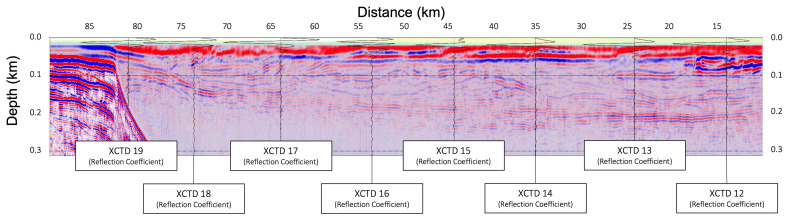
The seismic oceanographic image of BF10 for shallow water depths (under 300 m) with the reflection coefficient profiles for each XCTD station.

**Figure 7 sensors-23-06622-f007:**
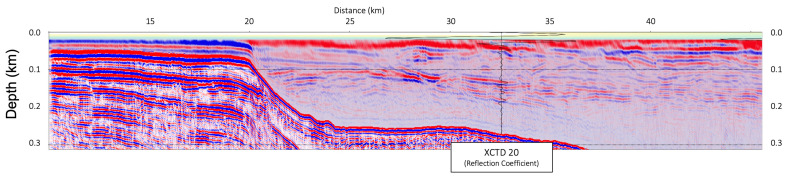
The seismic oceanographic image of BF11 for shallow water depths (under 300 m) with the reflection coefficient profile for the XCTD20 station.

**Figure 8 sensors-23-06622-f008:**
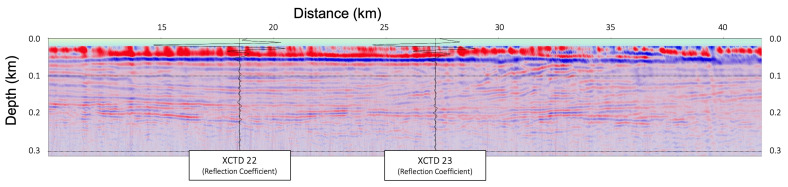
The seismic oceanographic image of BF12 for shallow water depths (under 300 m) with the reflection coefficient profiles for stations XCTD22 and 23.

**Table 1 sensors-23-06622-t001:** Parameters of the multichannel seismic survey during the Arctic expedition of the IBRV Araon in 2017 (ARA08C) on the Canadian Beaufort shelf, Arctic Ocean [14].

Acquisition Parameters	Values
Source	Two G.I. guns (210 cubic inches in total)
Shot interval	25 m
Group interval	12.5 m
Channel number	120
Minimum offset	85 m
Source & receiver depth	6 m
Sampling rate	1 ms
Total recording length	8 s

**Table 2 sensors-23-06622-t002:** Information on the XCTD measurements during the Arctic expedition of the IBRV Araon in 2017 on the Canadian Beaufort shelf, Arctic Ocean [14].

Station	Date	Time (UTC)	Longitude	Latitude	Depth
XCTD01	2 September 2017	4:07	138°19.3009′ W	69°41.5032′ N	148 m
XCTD02	2 September 2017	7:28	138°36.0697′ W	69°54.2250′ N	222 m
XCTD03	2 September 2017	10:07	138°50.1197′ W	70°04.5547′ N	316 m
XCTD04	2 September 2017	11:48	138°59.3782′ W	70°11.5034′ N	388 m
XCTD05	2 September 2017	14:05	138°44.9444′ W	70°11.8406′ N	383 m
XCTD06	2 September 2017	16:01	138°21.0126′ W	70°08.4472′ N	255 m
XCTD07	3 September 2017	9:46	139°33.7904′ W	70°09.1087′ N	202 m
XCTD08	3 September 2017	12:33	139°28.2543′ W	70°20.9371′ N	607 m
XCTD09	3 September 2017	14:50	139°23.8467′ W	70°30.2425′ N	785 m
XCTD10	3 September 2017	16:13	139°20.8336′ W	70°36.4878′ N	1250 m
XCTD11	3 September 2017	18:24	139°15.9515′ W	70°46.4286′ N	1741 m
XCTD12	3 September 2017	20:47	139°31.1775′ W	70°47.1764′ N	1805 m
XCTD13	3 September 2017	23:15	139°34.8686′ W	70°41.5127′ N	1705 m
XCTD14	3 September 2017	23:23	139°38.6842′ W	70°35.9066′ N	1233 m
XCTD15	4 September 2017	0:28	139°41.4555′ W	70°31.5702′ N	782 m
XCTD16	4 September 2017	1:34	139°44.9153′ W	70°26.3360′ N	671 m
XCTD17	4 September 2017	2:51	139°47.2517′ W	70°23.3024′ N	480 m
XCTD18	4 September 2017	4:05	139°51.5278′ W	70°15.8214′ N	375 m
XCTD19	4 September 2017	4:58	139°54.1667′ W	70°12.0187′ N	188 m
XCTD20	4 September 2017	9:29	139°25.1632′ W	70°11.6801′ N	280 m
XCTD21	4 September 2017	11:47	139°01.4251′ W	70°16.7027′ N	470 m
XCTD22	4 September 2017	12:53	138°58.1071′ W	70°21.5499′ N	607 m
XCTD23	4 September 2017	13:58	138°55.0283′ W	70°26.1055′ N	720 m

## Data Availability

Not applicable.

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
