# Peer review of "Frequency-Domain Reverse-Time Migration with Analytic Green’s Function for the Seismic Imaging of Shallow Water Column Structures in the Arctic Ocean"

_sensors, 2023, doi:10.3390/s23146622_

Round 1

Reviewer 1 Report

General comments

The manuscript proposes a new formulation of the frequency-domain reverse-time migration method for seismic oceanography based on the analytic Green's function and field data imaging performance has proved the correctness of the proposed method. I think this paper has met the requirements for publication, but there are still some details that need to be addressed, see the specified comments.

Specified comments

1. On line 28 of page 1, has verified should be modified as has been verified.

2. On line 95 of page 2, layer should be modified as layers.

3. Some of the formula characters are italic, but some are not, such as u in equation 3 of page 3. I suggest that the authors should unify the font of all formula characters  for italic.

4. A complete sentence should have a period and the authors should add a period at the end of the sentences such as line 140 of page 4.

5. The figure serial number is not written in the right way. For example, “Figure. 3.” on line 277 of page 10 should be modified as “Figure 3.”. The same problem occurs in the subsequent figures as well.

Minor editing of English language required

Author Response

Comments & Answer

General comments: The manuscript proposes a new formulation of the frequency-domain reverse-time migration method for seismic oceanography based on the analytic Green's function and field data imaging performance has proved the correctness of the proposed method. I think this paper has met the requirements for publication, but there are still some details that need to be addressed, see the specified comments.

Answer: Following your suggestion, our manuscript has been modified. Please confirm our revised manuscript. Thank you for your comments.

Specified comments:

1. On line 28 of page 1, "has verified" should be modified as "has been verified".

Answer: The correction has been made following your suggestions. On line 28 of page 1, we added "been" in our revised manuscript.

 2. On line 95 of page 2, "layer" should be modified as "layers".

Answer: In our revised manuscript, "layer" on line 95 of page 2 has been changed to" layers." Please confirm the revised manuscript.

3. Some of the formula characters are italic, but some are not, such as "u" in equation 3 of page3.

Answer: Following your direction, the formula character "u" in equation 3 has been changed to italic "u."

4. I suggest that the authors should unify the font of all formula characters  for italic.

Answer: We appreciate your careful review and suggestion. All formula characters have been changed to italic following your suggestions. Please confirm the revised manuscript.

5. A complete sentence should have a period and the authors should add a period at the end of the sentences such as line 140 of page 4.

Answer: The correction has been made following your direction. Please confirm the revised manuscript.

6. The figure serial number is not written in the right way. For example, "Figure. 3." on line 277 of page 10 should be modified as "Figure 3.". The same problem occurs in the subsequent figures as well.

Answer: The correction has been made following your directions. Please confirm the revised manuscript. Following your point out, Figure. 3. Other mistypes ("Figure. 4." and others) related to the figure serial number have also been corrected in our revised manuscript. Thank you for your suggestion.

Reviewer 2 Report

Review of the manuscript Frequency-domain Reverse-time Migration with analytic green’s function for the seismic imaging of shallow water column structures in the arctic ocean by Kang and Jang.

General comment

This manuscript is a well-written and nicely illustrated example of the use of high-quality marine seismic techniques to study the environmental phenomena, in this case, the Arctic Ocean temperature lineation. The subject is not new, but still rare. The article proposes also a simplification of the Reverse Time Migration technique with some significant assumptions. In my opinion manuscript in its current form does not convince the reader that the method is better comparing to full RTM or standard imaging methods.

Main problems

1) Why use RTM (an advanced and costly method for complicated structures) to study simple and regular layers in water? Yes, the contrasts are small, but RTM is really complicated/expensive method for complicated structures.

2) Line 70 – the largest simplification of the work, constant velocity in water. You are studying the thermocline structures and velocity is strongly correlated with temperature!

Line 142: again, assumption of constant velocity. This work is focused on shallow layers in water where the temperature variations are the largest and have the strongest effect on seismic imaging below them.

3) Line 172: these shallow water bubble oscillations are not as easy to remove, as the source signature strongly depends on the angle (due to both ghosting and seismic array configuration). The minimum offset is 85 meters, so directivity will be a significant problem

4) In the results FIG 3, 4 and 5 I see excellent images obtained by RTM, but I cannot see the state of the data before the processing. How it looked like without RTM, but with standard imaging. How can I evaluate the usefulness of the RTM in this case.?

5) I don’t see a reason to show 5c, 6a, 7a, 8a panels. These are really nice images of the structures under the water, but they have no meaning for this paper. The temperature profiles are presented in other panels and are clear.

6) Where is the Discussion, which is the most important part of each research paper? Showing the results of a new method, without comparison to current or easier/faster/cheaper methods, and claiming (line 360 From the numerical test we confirm that method can be used) is not enough to meet criteria of scientific discussion. Please extend this. In the conclusions, authors claim that the method is better/faster/less memory intensive and they are probably right, but this manuscript does not prove it. I would like to see the results of standard imaging and full RTM for comparison. Then the authors should qualitatively assess the quality of those results and compare the computational costs. Such discussion would make this article much stronger.

Small suggestions:

Line 25: mess -> mass

Line161: method can generate profiles -> method can generate useful results

Line 266: mess -> mass

Line 295: I don’t understand 90 L-km

Figure 5 is huge, and in current version printed in two pages. It have to be reorganize to match one page

My recommendation

I like the idea, and I confirm the quality of the data, text and figures is really good. I miss the arguments to support the author's claims. I suggest a major revision according to above-mentioned points that, in my opinion, will significantly improve this manuscript.

Author Response

Comments & Answer

General comment: This manuscript is a well-written and nicely illustrated example of the use of high-quality marine seismic techniques to study the environmental phenomena, in this case, the Arctic Ocean temperature lineation. The subject is not new, but still rare. The article proposes also a simplification of the Reverse Time Migration technique with some significant assumptions. In my opinion manuscript in its current form does not convince the reader that the method is better comparing to full RTM or standard imaging methods.

Main problems:

1) Why use RTM (an advanced and costly method for complicated structures) to study simple and regular layers in water? Yes, the contrasts are small, but RTM is really complicated/expensive method for complicated structures.

Answer: As you mentioned above, frequency-domain reverse time migration (RTM) is an advanced seismic imaging method complicated and expensive because of matrix factorization to calculate the forward and backward wavefield. The great advantage of RTM is that it can generate accurate seismic images from multi-channel seismic data. We've chosen the RTM method because we want an accurate seismic image for the water column structures at shallow water depths and our proposed RTM does not require the preprocessing procedure such as de-ghosting and de-bubble steps. In this study, we solve the problem of requiring significant computational resources of the RTM method by adopting the analytic Green's function instead of wave propagation modeling based on the finite element method. In our proposed RTM method, analytic Green’s function makes it possible to avoid a huge amount of computing procedure (FEM matrix factorization) and can handle with 3-dimension under the small workstation system. In the numerical test, the results (RTM images) show good correspondence with measured XCTD data; we can confirm that presenting structural information on water mass and boundaries of the shallow water column is well constructed, and RTM based on the analytic Green's function can be used to image the water column structures.

2) Line 70 – the largest simplification of the work, constant velocity in water. You are studying the thermocline structures and velocity is strongly correlated with temperature!

Line 142: again, assumption of constant velocity. This work is focused on shallow layers in water where the temperature variations are the largest and have the strongest effect on seismic imaging below them.

Answer: As you know, our proposed method assumes that the P-wave velocity and density of the water column are constant. In our manuscript, figure 2 shows the wave propagation travel time difference for the water column between the field-measured sound velocity and constant P-wave velocities (1,420, 1,425, 1,430, 1435, and 1440 m/sec). Following this figure, we note that the travel time difference is too small, especially in shallow depths under 750 meters. It means that the P-wave velocity of the water column can be assumed as the constant value for imaging the water column structures in shallow water depth. So, we insist that the variance in velocity of the water layer is small enough and can be assumed as the constant value in the RTM and seismic imaging field in our case. So, we note that the assumption of constant velocity doesn’t matter in the case of the frequency domain reverse time migration for constructing the water column image in the shallow water depth.

 3) Line 172: these shallow water bubble oscillations are not as easy to remove, as the source signature strongly depends on the angle (due to both ghosting and seismic array configuration). The minimum offset is 85 meters, so directivity will be a significant problem

Answer: As you mention, water bubble oscillations are a big problem in this shallow part. It is hard to remove. However, our proposed RTM method contains the source estimation algorithm, which works in de-bubble and de-ghosting steps; we confirm that our source estimation algorithm works well in our RTM method and presents reasonable seismic oceanographic images.

4) In the results FIG 3, 4 and 5 I see excellent images obtained by RTM, but I cannot see the state of the data before the processing. How it looked like without RTM, but with standard imaging. How can I evaluate the usefulness of the RTM in this case.?

Answer: To get a seismic image, acquired multi-channel seismic data must be performed in preprocessing steps. The Preprocessing procedures such as de-ghosting and de-bubble methods were required for acquired multi-channel seismic data, especially since getting a bubble-free dataset is not easy. As you know, the bubble is a big obstacle in seismic oceanographic images in the case of shallow water depth. Therefore, getting an interpretable seismic image by standard seismic imaging process is not easy and not guaranteed, especially for the shallow water depth. Because of that reason, we do not perform the conventional standard data processing for getting a seismic oceanographic image, especially in shallow water depth. However, our proposed RTM method includes a de-bubble algorithm in the source estimation step as written in the manuscript. It is another reason why we choose the RTM method for seismic oceanography.

5) I don’t see a reason to show 5c, 6a, 7a, 8a panels. These are really nice images of the structures under the water, but they have no meaning for this paper. The temperature profiles are presented in other panels and are clear. 

Answer: Our intent for presenting Fig 5c, 6a, 7a, and 8a is that the seismic events constructed near the 200-meter depth are related to the temperature change (increasing temperature). In these figures, we confirm that the seismic reflection event near the 200-meter depth means the boundaries between the Pacific and Atlantic waters in the Arctic circle.

6) Where is the Discussion, which is the most important part of each research paper? Showing the results of a new method, without comparison to current or easier/faster/cheaper methods, and claiming (line 360 From the numerical test we confirm that method can be used) is not enough to meet criteria of scientific discussion. Please extend this. In the conclusions, authors claim that the method is better/faster/less memory intensive and they are probably right, but this manuscript does not prove it. I would like to see the results of standard imaging and full RTM for comparison. Then the authors should qualitatively assess the quality of those results and compare the computational costs. Such discussion would make this article much stronger.

Answer: Thank you for your comments. I agree with your opinion that our method needs to compare to conventional frequency-domain RTM (full RTM) or standard imaging techniques to verify our proposed method and for readers' convenience. As you know, our proposed RTM method is based on the analytic Green's function, and it was developed to consider the seismic wave propagation in three-dimension. In the case of the conventional frequency domain RTM, the wave propagation modeling is required to calculate the forward and backward wavefield; we must solve the matrix of the wave equation of the FEM for the 3 dimensions. The seismic oceanography requires a high-frequency band (around 100 Hz) with a meter-scaled grid for water column structure imaging, and it cannot be possible to FEM matrix factorize and construct migrated images by conventional frequency domain RTM under the limitational computing resources. So, there is no way to compare each RTM method's accuracy and computational efficiency directly. Our proposed RTM method is the only way to construct the RTM seismic image with a high-frequency component (almost full-frequency band) of the water column.

Compared to standard imaging methods, preprocessing procedures such as de-ghosting and de-bubble methods were required for acquired multi-channel seismic data, especially since getting a bubble-free dataset is difficult. As you know, the bubble is the big obstacle in seismic oceanographic images in the case of shallow water depth. Therefore, getting an interpretable seismic image by standard seismic imaging process is not easy and not guaranteed, especially for the shallow water depth.

Small suggestions:

Line 25: mess -> mass

Answer: Following your direction, “mess” was changed to “mass.”

Line161: method can generate profiles -> method can generate useful results

Answer: Following your direction, the “method can generate profiles” was changed to the “method can generate useful results” in the revised manuscript.

Line 266: mess -> mass

Answer: Following your direction, “mess” was changed to “mass” in the revised manuscript.

Line 295: I don’t understand 90 L-km

Answer: L-km is line kilometers; it can be used to describe the length of survey tracks in the fieldwork. For the reader’s convenience, L-km was changed to km in the revised manuscript.

Figure 5 is huge, and in current version printed in two pages. It have to be reorganize to match one page

Answer: In the revised manuscript, figure resizing has been performed for the reader’s convenience.

Reviewer 3 Report

The submitted manuscript is devoted to a promising topic regarding the new algorithm for the seismic imaging of shallow water column. The article contains a theoretical substantiation of the algorithm and a sufficient amount of demonstration of the calculation results. Empirical material for modeling verification is the unique data of the polar marine expedition.

The Results and Discussion section clearly demonstrates the applicability of the method. Nevertheless, it remains unclear to what extent the presented new algorithm has an advantage over more conventional classic RTM method. A comparative analysis of the classical method and the new method has not been performed either by the criterion of better fit or by the criterion of lower computational cost.  

This seems to be a significant disadvantage requiring additional calculations. Especially in view of the fact that the introduction and conclusions affirm the great efficiency of the new method compared to the classical ones.

There are also minor typos and incorrect wording. For example:

line 51: «NMO» - explain the abbreviation at the first mention;

line 295: «L-km»;

line 314: «… with high nutrients with fresh (?)»;

lines 403-404: «…the analytic Green's function can be replaced with the wave propagation modeling…» - may be the other way around?

Author Response

Comments & Answer

The submitted manuscript is devoted to a promising topic regarding the new algorithm for the seismic imaging of shallow water column. The article contains a theoretical substantiation of the algorithm and a sufficient amount of demonstration of the calculation results. Empirical material for modeling verification is the unique data of the polar marine expedition. 

The Results and Discussion section clearly demonstrates the applicability of the method. Nevertheless, it remains unclear to what extent the presented new algorithm has an advantage over more conventional classic RTM method. A comparative analysis of the classical method and the new method has not been performed either by the criterion of better fit or by the criterion of lower computational cost.  

This seems to be a significant disadvantage requiring additional calculations. Especially in view of the fact that the introduction and conclusions affirm the great efficiency of the new method compared to the classical ones.

Answer: Thank you for your comments. I agree that our method needs to compare to conventional frequency-domain RTM to verify our proposed method and for readers' convenience. As you know, our proposed RTM method is based on the analytic Green's function, and it was developed to consider the seismic wave propagation in three-dimension. In the case of the conventional frequency domain RTM, the wave propagation modeling is required to calculate the forward and backward wavefield, we must solve the matrix of the wave equation of the FEM for the 3-dimensions. The seismic oceanography requires a high-frequency band (around 100 Hz) with a meter-scaled grid for water column structure imaging, and it cannot be possible to FEM matrix factorize and construct migrated images by conventional frequency domain RTM under the limitational computing resources. So, there is no way to compare each RTM method's accuracy and computational efficiency directly. Our proposed RTM method is the only way to construct the RTM seismic image with a high-frequency component (almost full-frequency band) of the water column.

There are also minor typos and incorrect wording. For example:

line 51: «NMO» - explain the abbreviation at the first mention; 

Answer: "NMO" is an abbreviation of the "Normal moveout". NMO has been changed to "Normal moveout" in the revised manuscript for the reader's connivance. Please confirm that in our revised manuscript.

line 295: «L-km»; 

Answer: L-km is line kilometers; it can be used to describe the length of survey tracks in the fieldwork. For the reader's convenience, L-km was changed to km in the revised manuscript.

line 314: «… with high nutrients with fresh (?)»; 

Answer: In the revised manuscript, “with high nutrients with fresh” has been changed to “; salinity is under 33 PSU”.

lines 403-404: «…the analytic Green's function can be replaced with the wave propagation modeling…» - may be the other way around?

Answer: Thank you for your suggestion. Yes! it is a very big typo error in our conclusions. “the analytic Green's function can be replaced with the wave propagation modeling…” has been changed to “the wave propagation modeling in the Frequency domain RTM can be replaced with the analytic Green's function for…”. Thank you again for your point-out.

Round 2

Reviewer 2 Report

The authors corrected small typos and used minor suggestions but did not address the major problems of the article. Instead, they answered my suggestions in detail. I would prefer to see those explanations as a part of the discussion in the manuscript, so the reader could read and evaluate them themself.

I agree with the explanation about the assumption of constant water velocity and the lack of possibility to compare the results with full RTM. Please add those few sentences to the conclusions.

Still, I don’t see a reason to show 5c, 6a, 7a, 8a panels, that in my opinion are not needed in this paper.

I suggest to accept this manuscript after just a small modifications in the conclusions.

Author Response

Dear Reviewer2,

We are grateful for your insightful comments on our manuscript. Our second manuscript version was revised to reflect most of the suggestions.

Here is a point-by-point response to the reviewer's comments and concerns.

Comments & Answer

The authors corrected small typos and used minor suggestions but did not address the major problems of the article. Instead, they answered my suggestions in detail. I would prefer to see those explanations as a part of the discussion in the manuscript, so the reader could read and evaluate them themself.

Answer: I agree with you. Following your suggestion, our second revised manuscript has added a “Discussion” section describing the major problems you mentioned in the previous step.

Please confirm our revised manuscript.

I agree with the explanation about the assumption of constant water velocity and the lack of possibility to compare the results with full RTM. Please add those few sentences to the conclusions.

Answer: We add a few sentences to explain the assumption of constant velocity and the lack of possibility to compare the results with full RTM in the “conclusions”.

Please confirm our revised manuscript.

Still, I don’t see a reason to show 5c, 6a, 7a, 8a panels, that in my opinion are not needed in this paper.

Answer: Following your suggestion, figures 5c, 6a, 7a, and 8a were removed with related texts and descriptions in our second revised manuscript.

Please confirm our revised version of the manuscript.

I suggest to accept this manuscript after just a small modifications in the conclusions.

Reviewer 3 Report

In the absence of the possibility of a comparative analysis of the proposed and conventional methods, this should be explained in the text. In addition, in this case, the statement in the conclusions that the proposed method "The method requires fewer computational resources..." is unfounded. It is recommended that this wording be removed.

Author Response

Dear Reviewer3,

We are grateful for your insightful comments on our manuscript. Our second manuscript version was revised to reflect most of the suggestions.

 Here is a point-by-point response to the reviewer's comments and concerns.

Comments & Answer

In the absence of the possibility of a comparative analysis of the proposed and conventional methods, this should be explained in the text.

Answer: We have added a new "Discussion" section to our manuscript, providing a detailed explanation for why we could not conduct a comparative analysis between the proposed and conventional methods in the revised manuscript. We kindly request your confirmation on the revised version of our manuscript.

In addition, in this case, the statement in the conclusions that the proposed method "The method requires fewer computational resources..." is unfounded. It is recommended that this wording be removed.

Answer: Thank you for your comments. I agree with you. Following your suggestion, the statement in the conclusions that the proposed method "The method requires fewer computational resources..." has been removed. Please confirm our revised manuscript.

 We appreciate your effort to detailed review and comment.
